# ATTRIBUTE-GUIDED IMAGE GENERATION WITH CAUSALLY DISENTANGLED REPRESENTATION

## ABSTRACT

Controllable image generation is a fundamental problem in machine learning and computer vision. Attribute-guided generative models enable explicit control over image content using labeled attributes, but often struggle to disentangle individual attributes and mitigate unwanted correlations—for example, adding eyeglasses may inadvertently alter a person's perceived age. In this work, we propose a novel attribute-guided generative framework designed to address these challenges. Our method learns a mask-based representation for each attribute label, encouraging disentanglement by limiting each attribute's influence to a small subset of the representation dimensions, while still preserving the information necessary to represent the label. To address attribute correlations, we incorporate classical causal discovery techniques to model inter-attribute dependencies and introduce a causal conditioning strategy that explicitly reduces undesirable correlations. Importantly, we provide theoretical guarantees showing that our method can recover the latent generative factors associated with individual attributes. Extensive experiments on diverse datasets demonstrate that our framework substantially improves attribute-level controllability and interpretability, outperforming existing baselines on attribute-guided image generation tasks.

## 1 INTRODUCTION

Controllable image generation has become a central focus in generative modeling, enabling applications such as photo editing, creative content design, and data augmentation. Unlike traditional generative models that produce arbitrary images from random noise, controllable models guide the generation process using high-level semantic cues, allowing outputs to better reflect user intent.

Conditional generative models enable such control by conditioning image synthesis on auxiliary inputs such as class labels (Karras et al., 2020; 2022; Dhariwal & Nichol, 2021; Peebles & Xie, 2023), textual prompts (Rombach et al., 2022; Esser et al., 2024; Labs, 2024b; Betker et al., 2023; Ramesh et al., 2022; Saharia et al., 2022), or structural signals like depth maps (Zhang et al., 2023; Zhao et al., 2023). In particular, text-to-image generation has advanced rapidly in recent years, driven by large-scale training and powerful diffusion-based architectures. Large-scale models such as Stable Diffusion (Rombach et al., 2022; Esser et al., 2024) have demonstrated impressive flexibility, visual quality, and semantic richness, generating diverse and realistic images directly from natural language prompts. These models currently represent the state of the art in open-ended conditional generation.

However, natural language can sometimes introduce ambiguity due to its contextual and subjective nature. Descriptions like "slightly smiling" or "partially bald" may be interpreted inconsistently across models or prompts, which can limit reliability in scenarios that require precise and repeatable control (see Table.3 and Fig.4). In this context, attribute-guided generation offers a complementary alternative by using explicit, structured supervision. Attribute-based models provide more consistent control over specific visual features, which is particularly useful in applications where interpretability, and fine-grained controllable generation are important.

Nonetheless, attribute-guided generative modeling introduces two critical challenges: **disentanglement** and **eliminating unwanted correlations**.

Firstly, given attribute labels, a key challenge is how to ensure the generative model learns disentangled representations for individual attributes. It is essential to ensure that modifications of one attribute

do not unintentionally influence others; for instance, changing a `background` attribute should not inadvertently alter other attributes like dog identity (see Fig. 5). Although recent advancements in Generative Adversarial Networks (GANs) (Hou et al., 2024; Dobler et al., 2022; Zhang et al., 2024), diffusion models (Yu et al., 2024; Peebles & Xie, 2023; Ma et al., 2024), and autoregressive models (Tian et al., 2024; Sun et al., 2024; Wu et al., 2024) have exhibited strong generative performances, these methods are primarily class-conditioned and are not explicitly optimized for fine-grained attribute disentanglement. Consequently, they offer limited support for precise, interpretable control over individual attributes. Several efforts have been made to generate images with specific attributes (Shen et al., 2020; Patashnik et al., 2021; Li et al., 2024). However, these approaches lack theoretical guarantees that the discovered directions in the latent space consistently correspond to the intended target attributes.

Secondly, attribute labels often exhibit inherent correlations within training datasets. For instance, attributes such as `eyeglasses` and `age` are frequently correlated, making independent generation challenging. Prior works, like InterfaceGAN (Shen et al., 2020), attempt to mitigate such correlations by projecting attribute vectors onto orthogonal directions. However, this approach presupposes independence between attribute subspaces and relies on linear classifiers, which may inaccurately reflect complex attribute correlations and inadvertently result in unwanted modifications (Chen et al., 2022) (see first column in Fig. 3). Thus, a significant question remains: How can generative models effectively isolate and eliminate these correlations to facilitate truly independent and robust attribute manipulation?

**Our Contribution.** In this paper, we propose an attribute-guided visual generative model that addresses two key challenges in controllable image generation: representation disentanglement and attribute correlation elimination. We begin by explicitly modeling the underlying data-generating process and introduce a mask-based representation learning method for attribute disentanglement. Specifically, we incorporate learnable masks into the representation learning process to ensure that each attribute label influences only a minimal subset of the representation dimensions, while still retaining all the information necessary to represent the label.

To further reduce unwanted correlations among attributes, we leverage classical causal discovery methods to infer the causal structure within the label space. Using this structure, we introduce a novel causal conditioning technique that learns attribute-specific representations conditioned only on their causal parents, thereby isolating each attribute's influence more effectively.

Importantly, we provide theoretical guarantees showing that our model can accurately recover latent factors uniquely associated with individual attributes. Extensive experiments on attribute-guided generation tasks demonstrate that our method significantly outperforms existing baselines in both controllability and visual quality.

## 2 RELATED WORK

**Causal Representation Learning** The goal of causal representation learning (CRL) is to reconstruct the true generation process of the data (Schölkopf et al., 2021). It can be viewed as a combination of disentanglement (Hyvärinen et al., 2023), representation learning (Bengio et al., 2013), and causal discovery (Spirtes et al., 2001; Glymour et al., 2019). Identifiability plays an important role in CRL. When the distributions of the estimation model and real data are matched, the learned latent variable is shown to be an invertible transformation of the true factor, i.e., the estimation only contains the information about the true factor (we call the variable *identifiable*). Learning concepts for compositional image generation is highly related to CRL as the identifiability helps achieve many desirable properties of the estimation model, such as robustness to outliers. For instance, CRL has been shown to be effective in many downstream classification tasks (Brehmer et al., 2022; Mitrovic et al., 2020; Wang et al., 2022; Lu et al., 2021). Recently, it has been shown that CRL can improve large language models Rajendran et al. (2024). In this paper, we present a framework for identifiable latnt variables and explore how CRL benefits attribute-guided image generation.

**Attribute-guided Generation and Controllable Generation.** Attribute-guided image generation is a key task in controllable image synthesis. Early methods build on conditional GANs (Mirza, 2014), generating images conditioned on class or attribute labels. Augmentation-aware GANs like DiffAug (Zhao et al., 2020), AugGAN (Hou et al., 2024), and ANDA (Zhang et al., 2024) improve training robustness by incorporating data augmentations and reducing label leakage. Causality-inspired

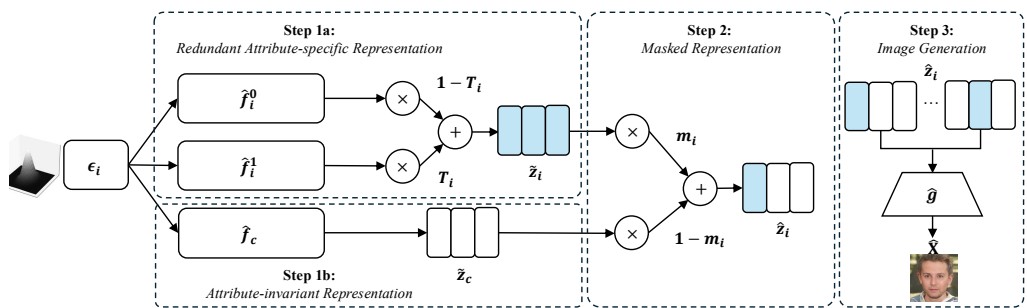

Figure 1: Overview of our mask-based framework for attribute-guided image generation (see Section 3.1 for details). Given noise samples $\epsilon$ from a prior distribution $\mathcal{N}(0, I)$, the model generates an attribute-invariant representation $\tilde{\mathbf{z}}_c$ and attribute-specific representations $\tilde{\mathbf{z}}_i$, which may contain redundant information. A learnable mask $\mathbf{m}_i$ is applied to modulate the contribution of each attribute $\mathbf{T}_i$, ensuring minimal influence from the attribute label $\mathbf{T}_i$ to inputs. All masked representations $\hat{\mathbf{z}}_i$ are combined and passed into the generator $\hat{g}$ to synthesize the output image $\hat{\mathbf{x}}$.

models such as CausalGAN (Kocaoglu et al., 2017) explicitly model label dependencies via causal graphs, but scale poorly with many attributes due to their need to generate full label distributions using auxiliary GANs. In contrast, our method leverages causal structure only for representation learning, avoiding this complexity. We compare our approach to CausalGAN both theoretically and empirically in the appendix. Latent-based approaches guide attribute control by manipulating latent representations. Some methods generate latents from attribute labels (Li et al., 2023; Suwała et al., 2024; Nie et al., 2021), while others learn attribute directions in latent space (Shen et al., 2020; Wu et al., 2021; Ling et al., 2021). InterfaceGAN (Shen et al., 2020), for instance, uses linear SVM boundaries for manipulation. In diffusion-based models, WPlus (Li et al., 2024) incorporates StyleGAN latents into the sampling process, while ConceptSlider (Gandikota et al., 2024) uses paired data to train LoRA-based adapters (Hu et al., 2022). Other methods enable attribute control via text prompts (Patashnik et al., 2021; Wei et al., 2023; Huang et al., 2023).

## 3 ATTRIBUTE-GUIDED CAUSAL IMAGE GENERATION

Given a set of images $\mathbf{x}$ and their corresponding attribute labels $\{\mathbf{T}_i\}_{i=1}^m$—obtained from pretrained attribute classifiers, advanced multimodal large models (Hurst et al., 2024; Yang et al., 2024), or in cases with weak or missing labels as discussed in Appendix B—our goal is to develop a generative model that produces high-fidelity images faithfully aligned with the specified attributes. An overview of the proposed approach is shown in Fig. 1.

To achieve this, we address two fundamental challenges:

**Problem 1: Learning disentangled representations.** To effectively model and manipulate multiple attributes, it is essential to learn disentangled representations for each attribute that isolate the underlying latent factors of variation specific to their corresponding labels. Without proper disentanglement, representations can become entangled, resulting in unintended changes to unrelated features. For example, modifying the shoe type from flat to heels may also change the color, as illustrated in the last row of Fig. 5 (first two columns).

**Problem 2: Mitigating unintended influence from correlated attributes.** Attribute labels in real-world datasets often exhibit causal or spurious correlations. These dependencies can cause changes in one attribute to inadvertently influence others. For example, the attributes **age** and **eyeglasses** are commonly correlated—older individuals are more likely to wear glasses. Consequently, setting the **eyeglasses** attribute from 0 to 1 may unintentionally alter the apparent age in the generated image (see Fig. 2(a)). This entanglement poses a major challenge for fine-grained, controllable image generation.

### 3.1 LEARNING DISENTANGLED REPRESENTATIONS VIA MASKING

To address both challenges—disentangling representations and mitigating unwanted correlations—it is essential to reconsider how attribute labels are incorporated into generative models. In particular,

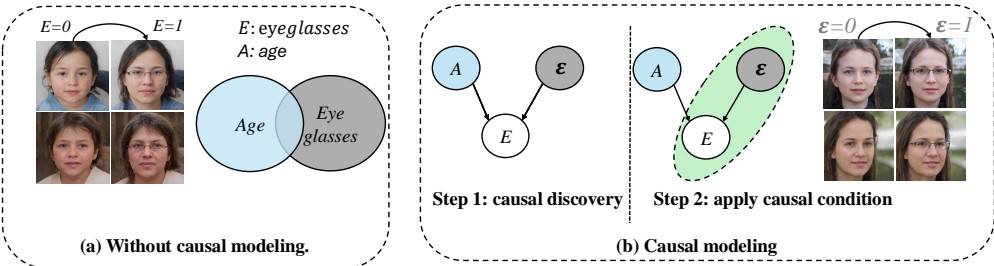

Figure 2: **The Necessity of Causal Modeling.** Modifying eyeglasses ($E$) without accounting for the causal relationship with age ($A$) can unintentionally change age-related features. By first discovering the causal direction between $A$ and $E$, and then applying causal conditioning, we can add eyeglasses without affecting perceived age (see Section 3.2).

we begin by analyzing why standard approaches to conditional generation often fail to produce properly disentangled representations.

A common approach to attribute-guided generation treats attribute labels as class indicators and trains a class-conditioned generative model. For example, given a random noise vector $\epsilon$ and an attribute label vector $\mathbf{T}$, one can train a conditional generative adversarial network (GAN) using the formulation $x = G(\epsilon, \mathbf{T})$, where $G$ is the generator. However, such models tend to learn entangled representations of the attributes in $\mathbf{T}$. As a result, modifying a single attribute may lead to unintended changes in other attributes (see Fig. 5). Moreover, these models offer no theoretical guarantees that the learned latent factors align with the true, independent sources of variation in the data.

**Data generating process.** To address these limitations, we aim to learn a separate latent representation for each attribute $\mathbf{T}_i$. Without proper constraints, many models can fit the data distribution but still produce entangled representations. To reduce this ambiguity, we assume the data is generated according to the following process:

$$\mathbf{z}_i := \mathbf{f}_i^1(\epsilon_i)\mathbf{T}_i + \mathbf{f}_i^0(\epsilon_i)(1 - \mathbf{T}_i);$$
$$\mathbf{x} := g(\mathbf{z}_c, \mathbf{z}_1, \mathbf{z}_2, \ldots, \mathbf{z}_m), \tag{1}$$

where $\epsilon_i$ is sampled from a prior distribution such as $\mathcal{N}(0, I)$, , and $\mathbf{f}_i^1$, $\mathbf{f}_i^0$ are transformation functions associated with the $i$-th label. In this formulation, $\mathbf{z}_c$ represents the attribute-invariant component shared across all labels (e.g., the orientation of a human face). Each $\mathbf{z}_i \in \mathbb{R}^{d_i}$ is the latent representation for attribute $\mathbf{T}_i \in \{0, 1\}$, with dimensionality $d_i$. For simplicity, we denote the complete latent code as $\mathbf{z} = (\mathbf{z}_c, \mathbf{z}_1, \mathbf{z}_2, \ldots, \mathbf{z}_m)$.

**Estimation model.** Informed by the underlying data generation process, we first construct an **attribute-invariant representation** corresponding to the true latent factor $\mathbf{z}_c$ from Eq. 1:

$$\tilde{\mathbf{z}}_c = \hat{\mathbf{f}}_c(\epsilon_i), \tag{2}$$

where $\hat{\mathbf{f}}_c$ is a shared function across all attribute labels. As a result, the output $\tilde{\mathbf{z}}_c$ carries no information about any specific attribute label $\mathbf{T}_i$ (step 1b in Fig. 1).

Next, we construct an **initial attribute-specific representation** for each label $\mathbf{T}_i$:

$$\tilde{\mathbf{z}}_i = \hat{\mathbf{f}}_i^1(\epsilon_i)\mathbf{T}_i + \hat{\mathbf{f}}_i^0(\epsilon_i)(1 - \mathbf{T}_i), \tag{3}$$

where $\tilde{\mathbf{z}}_i$ captures information specific to the attribute $\mathbf{T}_i$ (step 1a in Fig. 1). However, the amount of information each attribute conveys may vary significantly—for instance, the attribute `age` is likely to encode more complex features than `smile`. Moreover, if $\tilde{\mathbf{z}}_i$ has high dimensionality, it may inadvertently influence the representations learning of other attributes, such as $\tilde{\mathbf{z}}_j$ for $\mathbf{T}_j$, leading to undesired changes when only $\mathbf{T}_i$ is intended to be modified.

To address this, we propose a **mask-based estimation model**. Specifically, we introduce a learnable mask $\mathbf{m}_i$ for each attribute $\mathbf{T}_i$ and compute the final representation for attribute $\mathbf{T}_i$ as:

$$\hat{\mathbf{z}}_i = \mathbf{m}_i \odot \tilde{\mathbf{z}}_i + (1 - \mathbf{m}_i) \odot \tilde{\mathbf{z}}_c, \tag{4}$$

where $\odot$ denotes element-wise multiplication (step 2 in Fig. 1). This formulation enables us to control the degree to which each attribute affects the representation. For example, if $\mathbf{m}_i = \mathbf{1}$, then $\hat{\mathbf{z}}_i = \tilde{\mathbf{z}}_i$, meaning all elements of $\hat{\mathbf{z}}_i$ are influenced by attribute $\mathbf{T}_i$. Conversely, if $\mathbf{m}_i = \mathbf{0}$, then $\hat{\mathbf{z}}_i = \tilde{\mathbf{z}}_c$, which is attribute-invariant and ensures that modifying $\mathbf{T}_i$ has no effect on the output.

Finally, as shown in the final step of Fig. 1, we input all the masked representations into a generator $\tilde{g}$ to produce the output image $\hat{\mathbf{x}}$:

$$\hat{\mathbf{x}} = \tilde{g}(\tilde{\mathbf{z}}_c, \tilde{\mathbf{z}}_1, \tilde{\mathbf{z}}_2, \ldots, \tilde{\mathbf{z}}_m) \tag{5}$$
$$= \hat{g}(\hat{\mathbf{z}}_1, \hat{\mathbf{z}}_2, \ldots, \hat{\mathbf{z}}_m). \tag{6}$$

Following the conditional GAN framework, we employ a discriminator $D$ and perform adversarial training between the generator and discriminator to match the data distribution as:

$$\mathcal{L}_{\text{adv}} = \mathbb{E}_{(x,T) \sim p_{\text{data}}} \left[ \log D(x, T) \right] + \mathbb{E}_{z \sim p(z), \, T \sim p(T)} \left[ \log \left( 1 - D(\hat{x}, T) \right) \right]. \tag{7}$$

To prevent each attribute from exerting excessive influence, we apply $\ell_1$ sparsity regularization to the masks, encouraging each to affect only a small number of representation dimensions:

$$\mathcal{L}_{\text{sparsity}} = \sum_{i=1}^{m} \|\mathbf{m}_i\|_1. \tag{8}$$

**Full Objective.** Our overall objective for training the attribute-guided generative model is:

$$\mathcal{L}_{\text{full}} = \mathcal{L}_{\text{adv}} + \lambda_{\text{sparsity}} \mathcal{L}_{\text{sparsity}}, \tag{9}$$

where $\lambda_{\text{sparsity}}$ is a hyperparameter that balances the adversarial loss and the sparsity regularization.

## 3.2 REMOVING UNWANTED RELATION WITH CAUSAL MODELING

While our mask-based estimation model enables learning disentangled representations with independent attribute labels where $\mathbf{T_i} \perp \mathbf{T}_j \forall_{i,j}$, in practice, these labels are often *causally related*, which can lead to unintended side effects when modifying a single attribute. For example, although the attributes age $A$ and eyeglasses $E$ are causally linked, we may wish to modify eyeglasses without altering the perceived age. Without explicit causal modeling, such changes can inadvertently affect age, as illustrated on the left side of Fig. 2.

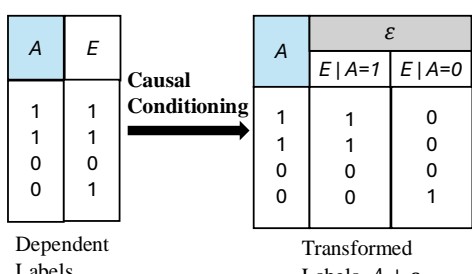

This problem arises because age (denoted $A$) is a *parent node* of eyeglasses (denoted $E$). According to causal reasoning principles (Pearl, 2009), intervening on a child node (e.g., modifying $E$) should not influence its parent $A$. Violating this principle often results in unnatural or implausible generations. Therefore, it is crucial to incorporate causal reasoning into the generative process—particularly when both precise attribute control and natural image generation are desired.

Table 1: **Example of Causal Conditioning**. After deciding the causal direction between $A$ and $E$, we use the two columns $E|A = 0, E|A = 1$ to act as surrogate of the unknown noise variable $\varepsilon$. Then we can change $\varepsilon$ to add eyeglasses without affecting $A$ since $A$ and $\varepsilon$ are independent.

To this end, we adopt a two-step strategy (right side of Fig. 2): first, we discover the causal structure among attributes; then, we modify the model's conditioning mechanism to respect this structure.

**Step 1: Discovering the causal structure and direction.** To ensure causally consistent generation, we first identify the causal relationships between attributes (e.g., determine whether $A \to E$ or $E \to A$). To reduce human efforts, we can apply automatic causal discovery algorithms such as the PC algorithm (Spirtes et al., 2001). If the output of the causal discovery method contains unoriented edges (as can occur with PC), we can resolve these using domain-specific background knowledge. The details of the causal discovery approach used in our experiments are provided in Appendix E.

**Step 2: Causal conditioning.** Suppose we have determined a causal relationship $A \to E$, represented by the structural causal model

$$E := f_E(A, \varepsilon),$$

where $\varepsilon$ is an exogenous noise variable independent of $A$, and $f_E$ is an unknown deterministic function. This formulation implies that $A$ and $E$ are dependent due to their shared causal mechanism.

To vary $E$ without altering $A$, we would ideally intervene on $\varepsilon$, as it directly influences $E$ and is independent of $A$. However, since $\varepsilon$ is unobserved, we approximate its influence by considering the conditional distributions $P(E \mid A = 0)$ and $P(E \mid A = 1)$. These conditionals capture the variability in $E$ that arises solely from $\varepsilon$ when $A$ is fixed. Therefore, they serve as surrogates for the unobserved noise: knowing $A$ and a realization from $P(E \mid A)$ is sufficient to determine $E$ through the structural equation $f_E$, and variation within each conditional reflects only the contribution of $\varepsilon$.

**Summary.** Based on this analysis, our causal modeling strategy proceeds as follows (illustrated in Fig. 2(b)). Given a set of attribute labels (e.g., left in Table 1), we first use causal discovery algorithms to infer the directionality of causal edges. Then, we replace each child node $E$ with its corresponding conditional variables $E|A = 0$ and $E|A = 1$. After this transformation, we can train a conditional generative model following the approach outlined in the previous section.

### 3.3 Identifiability Guarantee for the Underlying Latent Factors

In this section, we present identifiability guarantee for the underlying latent factors that correspond to the attribute labels. Specifically, we show that our estimation model recovers the true latent variables $\mathbf{z}_i$ up to an invertible transformation under some conditions. The key idea is to leverage sufficient variability across activating $\mathbf{f}_i^1$ and deactivating functions $\mathbf{f}_i^0$ to identify the true latent factors. The formal version and proof are in Appendix F.

**Theorem 1** (Identifying latent factors with a generative model (Informal)). *Consider the data generating process in Eq. equation 1. Suppose the noise distribution has a smooth positive density, and the generating, activation, and deactivation functions are smooth and invertible onto their images. Further assume that the activation and deactivation mechanisms $\mathbf{f}_i^1$ and $\mathbf{f}_i^0$ differ in a sufficiently strong way, in the sense that for $i = 1, \ldots, m$ and every $\epsilon_i$,*

$$\left( \frac{p_{\epsilon_i}}{(\mathbf{f}_i^1)'} \right)' \neq \left( \frac{p_{\epsilon_i}}{(\mathbf{f}_i^0)'} \right)',$$

*so that they cannot induce the same effect on the distribution. In addition, for each latent factor $\mathbf{z}_i$, there exist attribute labels $\mathbf{T}^{(k)}$ and $\mathbf{T}^{(l)}$ that differ only in the $i$-th entry, ensuring sufficient variation to isolate its effect. Then any generative model that assumes the same process, satisfies these conditions, and matches the observed distribution recovers the true latent variables $\mathbf{z}_i, i = 1, \ldots, m$ up to an invertible transformation for each attribute $\mathbf{T}_i$.*

This identifiability result holds for both independent and dependent attribute labels. In the dependent case, causal conditioning relies on the exogenous noise $\varepsilon$, independent of the target attribute, rather than on the parent label it depends on. Theorem 1 is central to disentangled representation learning for attribute-guided image generation. With the identifiability guarantee, when our estimation method matches the data likelihood, the model recovers the underlying latent factors, which are inherently disentangled. This property enables a variety of downstream tasks, including controllable image generation and smooth interpolation.

## 4 Experiments

In this section, we evaluate our proposed method. Section 4.1 describes the experimental setup, Section 4.2 presents results on datasets with independent attributes, and Section 4.3 reports results on datasets with causally related attributes.

### 4.1 Setup

**Datasets.** We evaluate our method on two types of datasets: those with *independent* attribute labels and those with *dependent* attribute labels. For datasets with *independent attributes*, we construct four settings. In **AFHQDog** (Choi et al., 2020), we use a vision-language model (Team, 2024) to annotate three binary attributes: whether the dog's background is green, whether its ears are perked,

| Method | AFHQDog | ZAPPOS | ColorMNIST | LSUNBedroom |
|---|---|---|---|---|
| StyleGAN2-ADA (Karras et al., 2020) | 10.03 | 42.95 | 110.56 | 28.06 |
| MultiClassGAN (Dobler et al., 2022) | 9.57 | 4.48 | 109.88 | 16.13 |
| AugGAN (Hou et al., 2024) | 13.73 | 4.45 | 17.79 | 10.30 |
| ANDA (Zhang et al., 2024) | 9.82 | 4.81 | 101.57 | 39.36 |
| Ours ($\lambda_{\text{sparsity}} = 0$) | 10.47 | 4.15 | 16.36 | 17.75 |
| Ours | **9.25** | **3.96** | **4.32** | **8.09** |

Table 2: **FID scores across multiple independent attribute-guided datasets**. Lower is better.

| Method | Eyeglasses | | | | |
|---|---|---|---|---|---|
| | DINO ↑ | ID ↑ | Acc ↑ | $L_1$ ↓ | Disen ↓ |
| InterfaceGAN | 0.85 | 0.61 | 0.63 | 0.11 | 4.19 |
| StyleCLIP | 0.88 | 0.63 | 0.90 | 0.07 | 3.31 |
| WPlus | 0.85 | 0.77 | 0.91 | 0.09 | 4.16 |
| ConceptSlider | 0.80 | 0.74 | 0.92 | 0.06 | 2.82 |
| SD3.5-Large | 0.60 | 0.43 | 1.00 | 0.19 | 4.66 |
| Flux.1.dev | 0.72 | 0.48 | 1.00 | 0.12 | 3.98 |
| SANA | 0.78 | 0.49 | 0.97 | 0.11 | 3.26 |
| **Ours** | **0.95** | **0.78** | **0.96** | **0.04** | **2.53** |

| Attr | Method | DINO ↑ | Acc ↑ | Disen ↓ |
|---|---|---|---|---|
| Beard | ConceptSlider | 0.90 | 0.67 | 3.15 |
| | SANA | 0.83 | 1.00 | 4.49 |
| | **Ours** | **0.93** | **1.00** | **2.86** |
| Chubby | ConceptSlider | 0.93 | 0.75 | 2.39 |
| | SANA | 0.80 | 0.84 | 4.77 |
| | **Ours** | **0.96** | **1.00** | **1.96** |
| Smile | ConceptSlider | 0.96 | 0.97 | 1.68 |
| | SANA | 0.87 | **1.00** | 4.30 |
| | **Ours** | **0.98** | 1.00 | **1.57** |

Table 3: **Results on Attribute Modification for Face Image Generation.**

and whether its mouth is open. In **ZAPPOS** (Yu & Grauman, 2014), we use two sets of attributes: (1) shoe type (flat vs. heels) and (2) category (sandals, slippers, or shoes). For **ColorMNIST**, we generate labels for three attributes: background color, digit thickness, and digit color. In **LSUN Bedroom** (Yu et al., 2015), we extract 10 binary room attributes using a visual question answering (VQA) model. For the case of *dependent attributes*, we use the **FFHQ** dataset (Karras et al., 2019), where attributes exhibit strong correlations. We apply a pretrained CELEBA classifier to annotate 37 facial attributes. The dataset is used at a resolution of $512 \times 512$.

**Implementation, Baselines, and Metrics.** We develop our method based on the StyleGAN2-ADA code (Karras et al., 2020). We provide our code and implementation details in the supplementary material. For independent attribute image generation evaluation, we benchmark our approach against state-of-the-art GAN methodologies, including StyleGAN2-ADA (Karras et al., 2020), MulticlassGAN (Dobler et al., 2022), AugGAN (Hou et al., 2024), and ANDA (Zhang et al., 2024). While we considered including diffusion models in our comparative analysis,[1] their computational requirements proved prohibitively expensive for our experimental framework. We use the standard Frechet Inception Distance (FID) to measure the divergence between generated data and training data. For causal-related attribute generation, we compare with InterfaceGAN (Shen et al., 2020), StyleCLIP (Patashnik et al., 2021), WPlus (Li et al., 2024), and ConceptSlider (Gandikota et al., 2024), SD3.5-Large (Esser et al., 2024), Flux.1.dev (Labs, 2024a), and SANA (Xie et al., 2024). We measure the performance with ArcFace ID (Deng et al., 2019) similarity between the generated paired faces, DINO (Caron et al., 2021) similarity, accuracy with pretrained classifier, and $L_1$ distance between the generated paired images. Following InterfaceGAN (Shen et al., 2020), we also perform a re-scoring disentanglement analysis, which evaluates how other attributes change when an edit is applied to a target attribute. We refer to this metric as Disen.

### 4.2 RESULTS WITH INDEPENDENT ATTRIBUTES

Table 2 reports FID scores on four benchmark datasets—AFHQDog, ZAPPOS, ColorMNIST, and LSUN—comparing our method (with and without sparsity regularization) against strong GAN baselines. Our full model consistently achieves the lowest FID across all datasets, demonstrating its effectiveness in generating high-quality, attribute-controlled images. We provide visual comparisons of generated samples in Appendix.A.

---

[1]training diffusion models on the AFHQ dataset at 64×64 resolution requires approximately 768 V100 GPU hours (Karras et al., 2022)

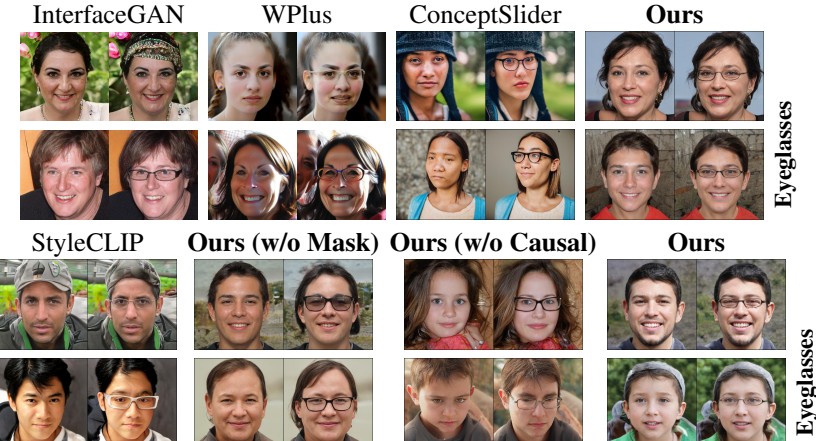

Figure 3: **Generation Results After Attribute Modification.** WPlus (Li et al., 2024) and ConceptSlider (Gandikota et al., 2024) fail to account for correlations between eyeglasses and other attributes, often producing entangled outputs, e.g., WPlus tends to increase the apparent age when adding eyeglasses. While InterfaceGAN (Shen et al., 2020) attempts to mitigate such correlations using orthogonal projection, its reliance on a linear classifier can yield inaccurate attribute directions. As a result, it may fail to correctly add eyeglasses and still affect age-related features. In contrast, our method modifies only the target attribute while preserving correlated features, thanks to our mask-based design and causal conditioning framework.

### 4.3 RESULTS WITH CAUSALLY-RELATED ATTRIBUTES

**Attribute-guided Generation as a Complement to Text-to-Image Models**  Recent text-to-image (T2I) models have demonstrated impressive performance. However, attribute-guided generation remains a valuable complement, particularly when precise and smooth generation or editing is required. As shown in Table 3, state-of-the-art T2I models achieve strong instruction-following capabilities (evidenced by high target edit accuracy). Nevertheless, even minor modifications to the text prompt (e.g., a single word) can cause drastic changes in the output, as reflected in the low DINO, ID, L1, and Disen scores. Furthermore, we prompted GPT to enumerate 10 variations of facial size and hair and then generated corresponding images. As illustrated in Fig. 4, even subtle changes in the prompt often lead to identity loss and unsmooth interpolations.

**Comparisons.** Quantitative results are reported in Table 3, and qualitative results are shown in Fig. 3. Our method achieves the best performance in attribute-controlled image generation, particularly in challenging cases with correlated attributes. For instance, the eyeglasses attribute is known to be strongly correlated with age (Shen et al., 2020). InterfaceGAN (Shen et al., 2020) attempts to mitigate this issue using orthogonal projection, but the linear boundary obtained by its classifier is often inaccurate, leading to unsatisfactory results (see Fig. 3). More recent methods such as WPlus (Li et al., 2024) and ConceptSlider (Gandikota et al., 2024), built on the Stable Diffusion v1.4 backbone, improve upon InterfaceGAN but still fail to account for attribute correlations. As a result, modifying one attribute (e.g., eyeglasses) often causes unintended changes in others, such as age or pose.

**Ablation Study** We also provide a qualitative ablation study on the effectiveness of the proposed sparsity constraint and causal modeling component (quantitative results are included in the supplementary material) in Fig. 3. Without the sparsity constraint, the eyeglasses attribute tends to affect too many input elements, resulting in numerous unintended changes in the generated images. Without causal modeling, although sparsity helps preserve some attribute disentanglement, modifying eyeglasses still causes undesirable effects—such as making the person appear older. In contrast, our full method consistently achieves better disentanglement, enabling precise control over individual attributes while minimizing unintended alterations.

**Smooth Interpolation** To thoroughly evaluate attribute control capabilities, we perform interpolation by gradually varying the attribute values, with results shown in Fig. 4. The strong baseline Concept-Slider (Gandikota et al., 2024) achieves interpolation by adjusting the corresponding LoRA scale. However, we find that ConceptSlider exhibits a sudden change issue: it either produces irrelevant

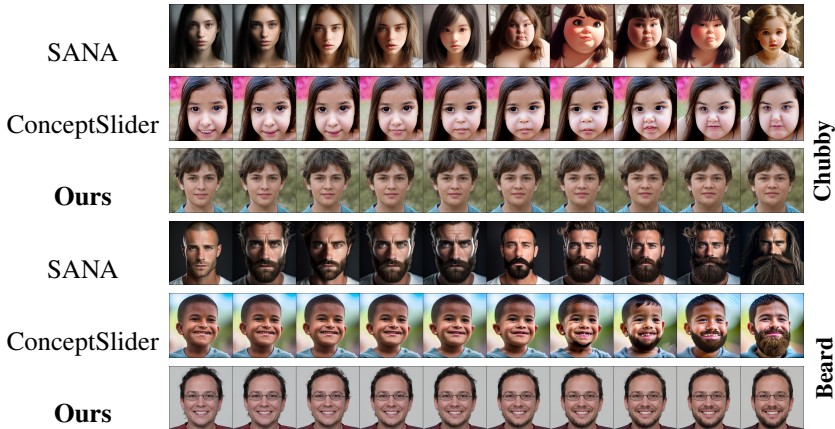

Figure 4: **Comparison of Interpolations for Attribute Control.** The baseline method (Gandikota et al., 2024) exhibits a sudden-change phenomenon: at low control strengths, it introduces unintended modifications to other attributes (e.g., changes to the mouth in the first row), followed by an abrupt activation of the target attribute accompanied by additional, undesired alterations. In contrast, our method achieves smooth and continuous interpolations, consistently modifying only the intended attribute throughout the process. More interpolation results are provided in the Appendix.A.

modifications at low scales or abruptly introduces the target attribute along with undesired changes. In contrast, our method progressively and consistently applies the target attribute as the value increases, without affecting other attributes in the output images. This highlights the superiority of our approach in scenarios where precise, fine-grained control is essential for attribute-guided image generation.

**Extension to text-to-image models**  Beyond the GAN framework, our general masking principles can also be applied to text-to-image models. As an example, we extend the Flux.1-dev model (Labs, 2024a) in a way that mirrors our GAN implementation. In GANs, we introduce attribute-specific and attribute-invariant representations, and apply a sparse mask to limit their influence. In diffusion models, these masks can be interpreted over timesteps: we begin with attribute-invariant prompts (e.g., *human face*) at early timesteps and switch to the attribute-specific prompt (e.g., *human face with eyeglasses*) at the minimal timestep necessary to ensure the attribute is generated. For causal conditioning, instead of relying on a single attribute prompt (e.g., *eyeglasses*), we condition on pairs of prompts (e.g., *young, eyeglasses* and *old, eyeglasses*), consistent with our GAN formulation. We find that these two extensions al-

| Method | Acc↑ | Disen↓ |
|---|---|---|
| Vanilla Finetune | 1.00 | 6.38 |
| +Time-mask | 0.94 | 4.81 |
| +Causal Modeling | 1.00 | 2.46 |

Table 4: Results on Flux.1-dev for adding eyeglasses.

ready yield substantial improvements over vanilla LoRA training. A limitation of the current T2I extension to the GAN framework is that it is fully controlled by text inputs, which results in a lack of support for smooth interpolation, similar to SANA (Xie et al., 2024) as shown in Fig. 4; more sophisticated methods may be explored in future work.

## 5  LIMITATIONS, DISCUSSION, AND CONCLUSION

Although our method achieves strong performance across various attribute-guided image generation tasks, it still faces certain limitations. One notable challenge is generating complex or uncommon compositions from limited training data. This difficulty is partly due to our reliance on StyleGAN2-ADA, which has known limitations in handling uncurated or compositionally complex data (Sauer et al., 2022). Addressing this issue remains an important direction for future work.

In this paper, we proposed an attribute-guided image generation framework that tackles two main challenges: disentangling attribute-specific representations and reducing the impact of correlated attributes. Our method is based on a principled data-generating assumption and uses a mask-based mechanism to limit each attribute's influence. To handle label correlations, we incorporate causal discovery and introduce a causal conditioning strategy to remove unwanted dependencies. Experiments across multiple datasets show that our approach consistently outperforms existing baselines in controllability, disentanglement, and image quality.

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

# Appendix for "Attribute-Guided Image Generation with Causally Disentangled Representation"

**Use of large language models (LLMs).** We use LLMs only to detect typos and refine wording in the paper; they are not involved in the idealization process.

## SUMMARY OF THE APPENDIX

In this appendix, we provide additional details and results to support the main paper:

- Section A presents additional generated samples and interpolation results using the strong baseline ConceptSlider (Gandikota et al., 2024).
- Section B discusses how our method can be extended to handle scenarios where attribute labels are not available.
- Section C provides a more comprehensive overview of related work.
- Section D presents comparisons with CausalGAN (Kocaoglu et al., 2017), including both causal analysis and empirical evidence demonstrating that our method is more data-efficient and produces higher-quality interventional samples.
- Section E shows the full causal graph used in our approach.
- Section F provides the proof of the identifiability results stated in the main paper.
- Section G includes implementation details of our method. We also provide the training code in the supplementary files.
- Section H presents both qualitative and quantitative ablation results, further supporting the effectiveness of the proposed method.

## A  MORE EMPIRICAL RESULTS

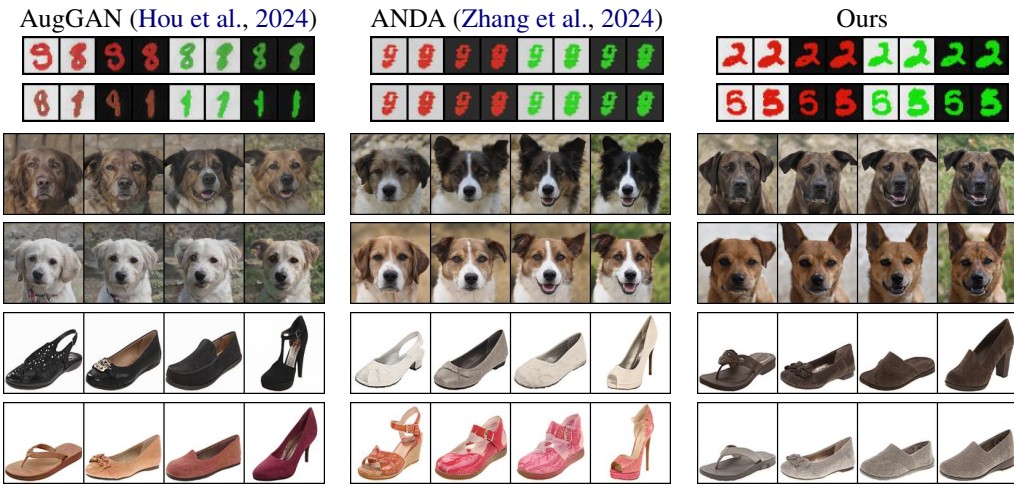

Figure 5: **Attribute-Guided Generation Results.** Each row illustrates sequential attribute modification using our method. (a) **MNIST**: progressive changes in digit thickness, background color, and foreground color. (b) **AFHQDog**: step-by-step addition of perked ears, an open mouth, and a green background. (c) **ZAPPOS**: transformation across footwear types—from sandals to shoes to slippers—with added heels. Baseline methods, including ANDA (Zhang et al., 2024) and AugGAN (Hou et al., 2024), often introduce unintended artifacts, such as color shifts during shoe transformations. In contrast, our method preserves precise attribute control, highlighting the effectiveness of our method.

**Comparison with ConceptSlider** We provide additional generated samples in Fig. 6. Interestingly, the baseline ConceptSlider (Gandikota et al., 2024) often fails to generate the intended attributes when faced with rare combinations. For example, it fails to add a beard to a female face, instead altering the lip color (first row). Moreover, when using the same strength parameter, ConceptSlider sometimes produces overly heavy beards or no beard at all, indicating that each sample may require its own carefully tuned hyperparameter. This inconsistency suggests that the baseline method is less reliable for controllable, attribute-guided generation. When attempting to generate chubby faces, it either changes the identity of the original subject (fifth row) or introduces irrelevant modifications, such as altering clothing color (fourth row). In contrast, our method successfully generates the target attributes—even in uncommon scenarios like beards on female faces—while preserving unrelated features and faithfully modifying only the intended attribute.

**Interpolation Comparison** We present the interpolated samples in Fig. 7. Similar to the results shown in the main paper, the baseline ConceptSlider (Gandikota et al., 2024) suffers from a sudden-change phenomenon, where the target attribute appears abruptly and is often accompanied by unintended modifications. For example, it suddenly adds a heavy beard to a female face while also altering the eyes (second row). In another case, it modifies unrelated attributes during interpolation—for instance, the clothing of the boy changes as the strength of the "smile" attribute increases (ninth row). In contrast, our method achieves smooth and gradual interpolation between the absence and presence of the target attribute, while preserving all other attribute values.

## B  LEARNING WITHOUT ATTRIBUTE LABELS

Our method, as introduced in the main paper, relies on access to attribute labels. With these labels, our model can disentangle the underlying concepts associated with each attribute. However, in real-world scenarios, such labels may not always be available. In this section, we propose solutions to address this limitation and demonstrate that our model can also *learn attribute labels jointly* when they are not provided.

First, we relax the requirement for detailed attribute labels and instead assume access to *class labels* only. We then introduce an auxiliary *labeler network* that takes the class label as input and predicts the corresponding attribute labels. This labeler network is trained jointly with our main model. To regularize the predictions and encourage sparsity—we apply an additional $\ell_1$ penalty to the estimated attribute labels. In other words, we expect the predicted attribute labels to be sparse.

We find that this simple design is effective on several datasets. As shown in Fig. 8, our method is capable of learning meaningful attribute labels, such as digit color and shape, directly from data and class labels. Furthermore, Fig. 9 demonstrates that our method can also discover distinct concepts in more challenging datasets—for example, learning both painting styles and subjects using only class labels and image data.

Second, we explore the most challenging setting, where only *image data* are available, and no labels (class or attribute) are provided. Since we have shown success in learning attribute labels with only class supervision, we propose extending our approach to the fully unsupervised case using *clustering techniques*. For example, (Liu et al., 2020) demonstrated that it is possible to cluster images during GAN training. We suggest adapting their method to cluster samples and estimate attribute labels in conjunction with our model. We leave this extension as a promising direction for future work.

## C  RELATED WORK

**Causally-aware Generative Models** CausalGAN (Kocaoglu et al., 2017) proposes to build a causal generative following the causal graph. For example, if $X \rightarrow Z \leftarrow Y$, then it employs three neural networks for each node and the input of neural network of $Z$ consists of the outputs of neural network for node $X$ and node $Y$. CausalVAE (Yang et al., 2021) assumes that there exists underlying causal structure among the latent variables and add a causal layer to learn such information. (Wen et al., 2022) employs CausalGAN for tabular data generation while supporting part of the causal graph. CGNN (Goudet et al., 2018) learns causal graph and the functions by minimizing the divergence between the generated data and real data. DECAF (Van Breugel et al., 2021) reconstructs each variable with its parents as conditioning and generates fair synthetic data. (Moraffah et al., 2020)

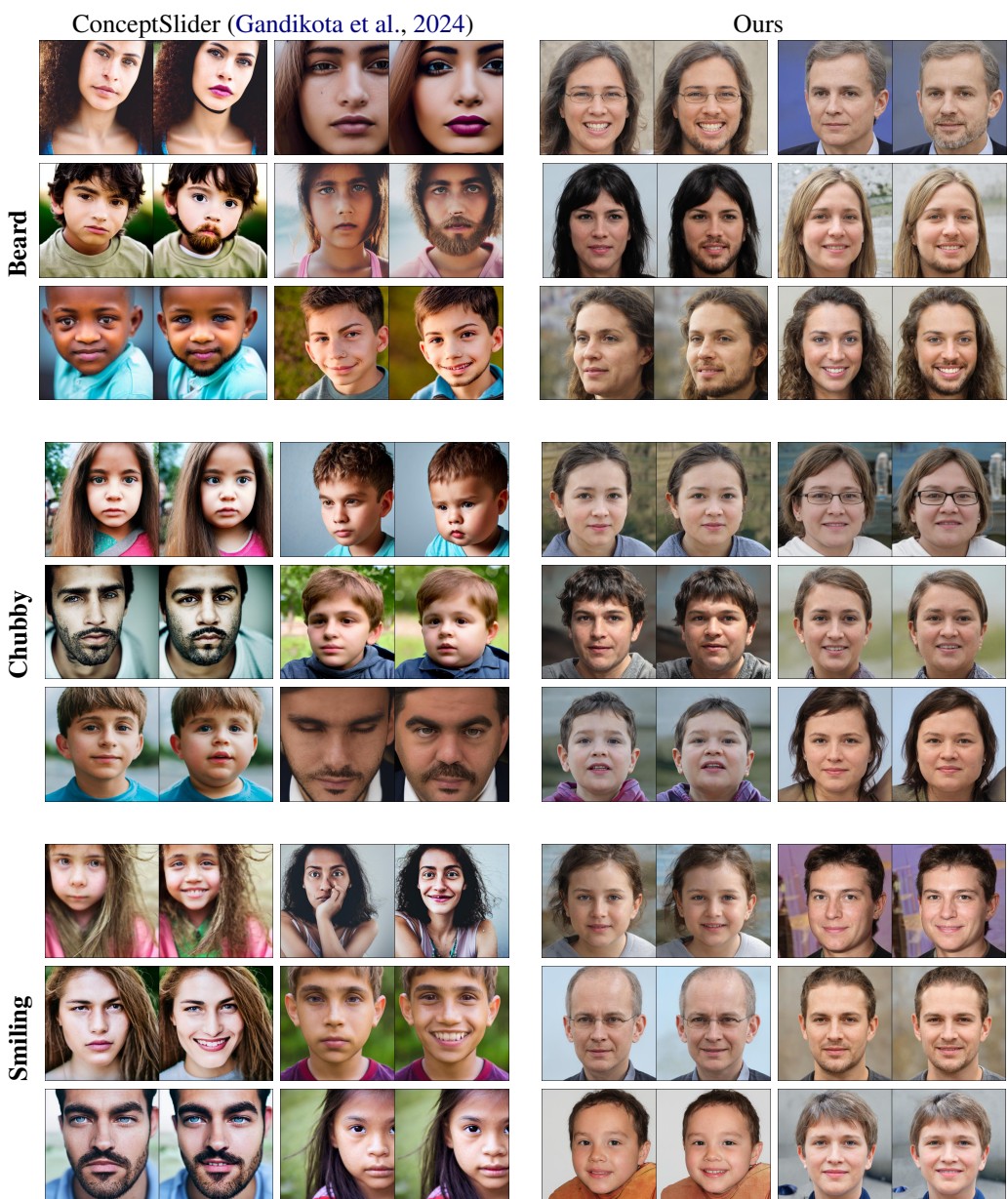

Figure 6: **Comparisons of Attribute-Guided Generation.** The baseline method, ConceptSlider (Gandikota et al., 2024), often fails to generate beards for uncommon attribute combinations, such as a girl with a beard. Additionally, we observe that ConceptSlider sometimes produces overly heavy beards or exaggerated smiles—even when using the same strength parameter—while in some cases it fails entirely. These inconsistencies suggest that the method is unreliable for controllable generation. In contrast, our approach accurately recovers the latent factors corresponding to the target attribute labels, enabling faithful modification of the desired attribute without affecting unrelated features.

assumes the Structural Causal Model (SCM) is linear and proposes to learn the causal graph with GAN. CNF (Javaloy et al., 2024) recovers the causal model with normaling flow given causal ordering information. CGN (Sauer & Geiger, 2021) assumes that images are generated by four components: shape, texture, background and composer and train a conditional GAN with corresponding labels. CAGE (Bose et al., 2022) examines the causal relationship between a pair of variables using potential outcome framework and generates counterfactual images.

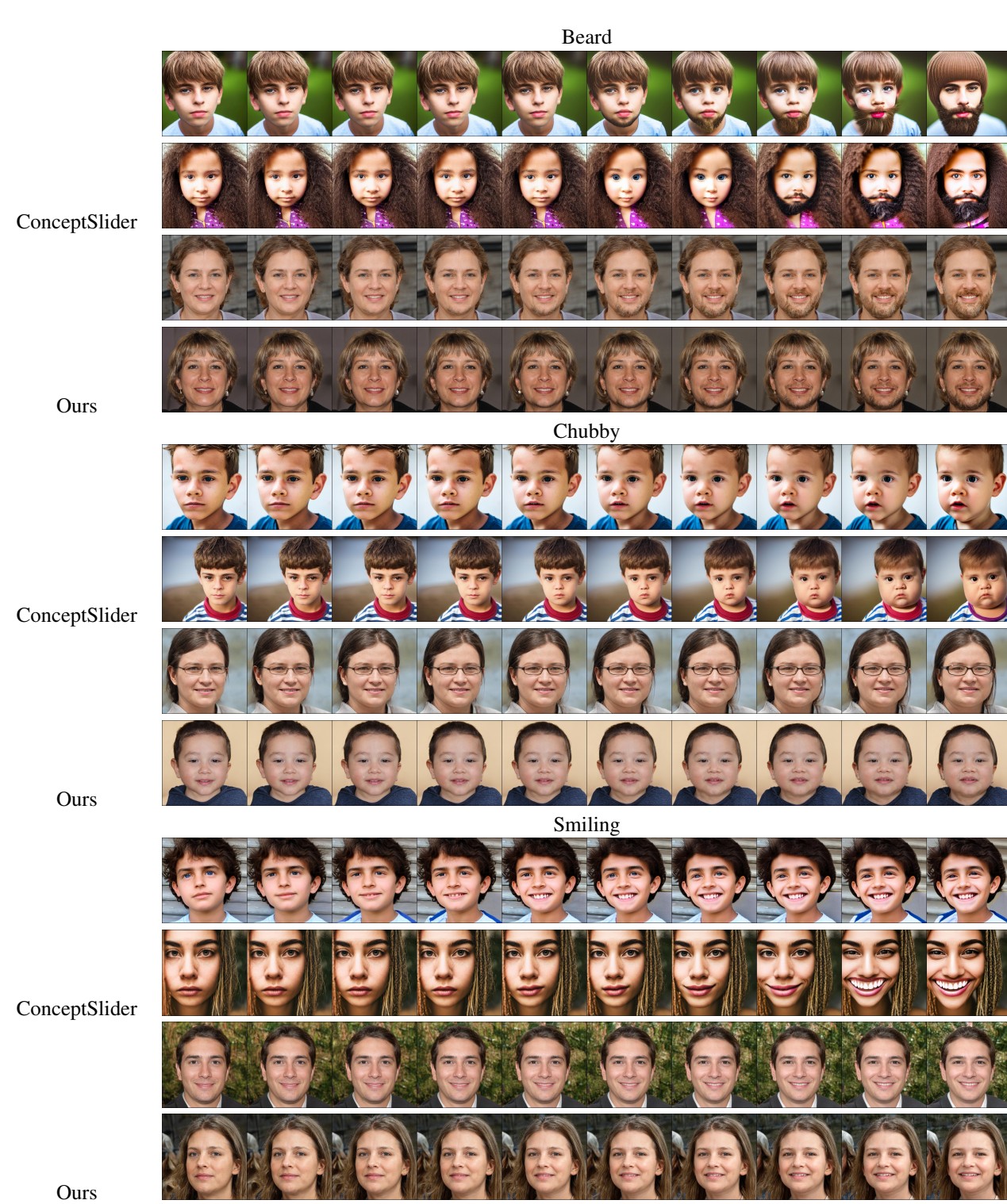

Figure 7: **Interpolation Comparisons.** The baseline method, ConceptSlider (Gandikota et al., 2024), exhibits a sudden-change phenomenon: at lower strength values, there is little to no effect, but beyond a certain threshold, it introduces abrupt and excessive changes—including alterations to unrelated attributes. In contrast, our method demonstrates more stable and controlled behavior, gradually modifying only the target attribute as the strength increases.

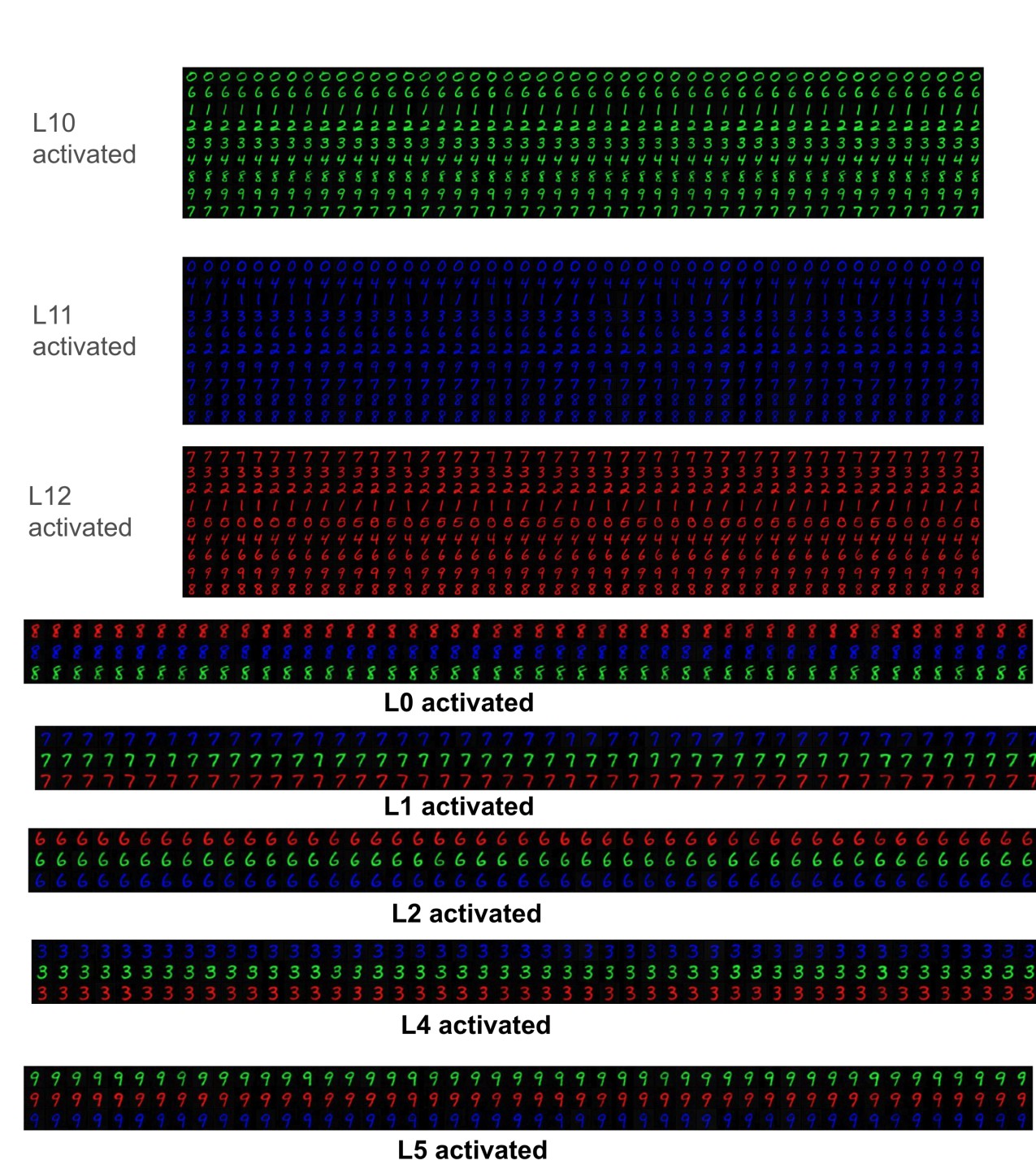

Figure 8: We can learn the attribute labels (denoted by $L$) jointly with our model **when attribute labels are not given**. On this MNIST dataset, we are able to learn the digit shape labels and digit colors from data.

$L_0$ activated

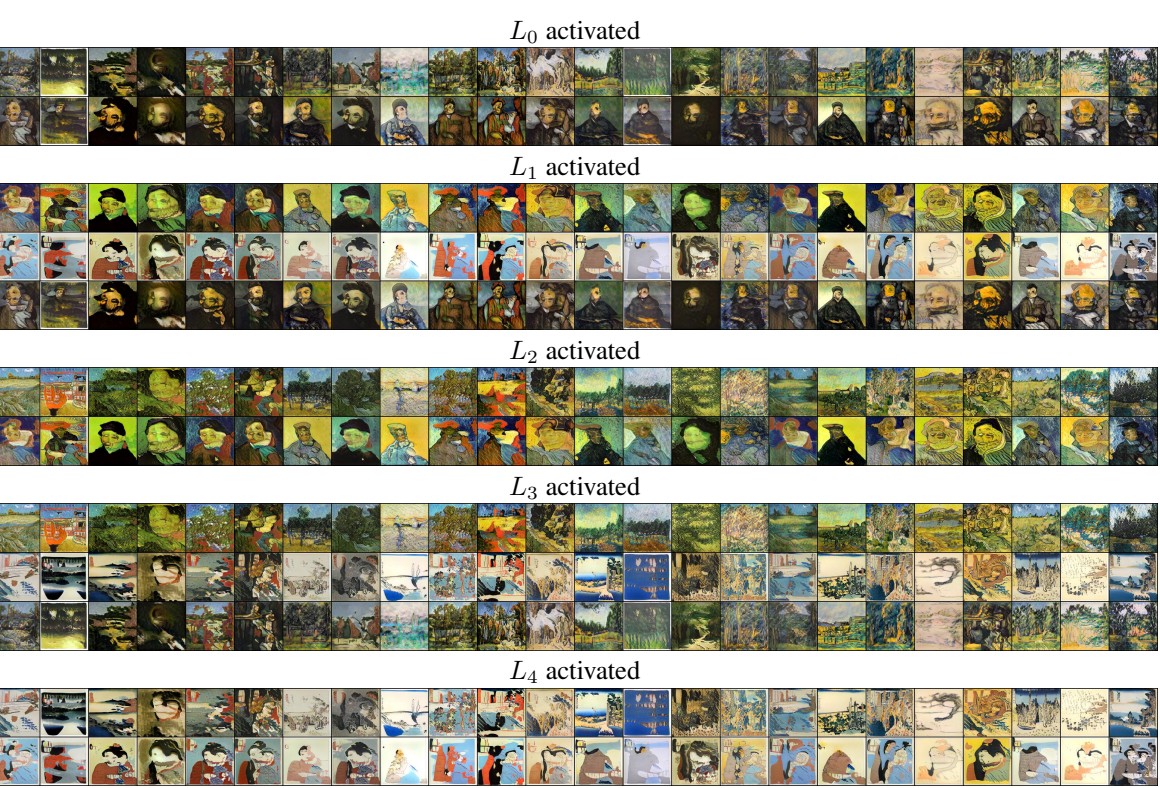

$L_1$ activated

$L_2$ activated

$L_3$ activated

$L_4$ activated

Figure 9: We can learn the attribute labels (denoted by $L$) jointly with our model **when attribute labels are not given**. On this artist dataset, we successfully disentangle different concepts, e.g., human face and painting style. For example, $L_1$ denotes the human face attribute and $L_2$ denotes the Van-Gogh painting style.

**Nonlinear ICA** Nonlinear ICA is a challenging ill-posed problem because the latent variables are generally not identifiable without any assumptions (Hyvärinen & Pajunen, 1999). Existing works resolve this issue by leveraging sufficient variability on the distribution of latent variables to obtain identifiability, where the distributions are indicated by auxiliary variables such as time indices and domain indices (Hyvarinen & Morioka, 2016; 2017; Hyvarinen et al., 2019; Khemakhem et al., 2020). Another line of works impose restrictions on the mixing function including certain function classes (Hyvärinen & Pajunen, 1999; Taleb & Jutten, 1999; Gresele et al., 2021; Buchholz et al., 2022) and sparse mixing function (Zheng et al., 2022).

## D  CAUSALGAN AND CAUSAL CONDITIONING

CausalGAN also assumes that there exists a causal relationship among the attribute labels. To capture this, it introduces an additional *causal controller* to model the joint distribution of the labels. Specifically, suppose a causal graph $X \rightarrow Q \leftarrow Y$. CausalGAN first generates $X$ and $Y$ using neural networks $x = f_x(z_x)$ and $y = f_y(z_y)$, respectively. It then generates $q = f_q(x, y, z_q)$, where $z_x$, $z_y$, and $z_q$ are latent variables. In other words, it requires using $n$ separate neural networks to model the causal generation process when there are $n$ attribute labels. Finally, all generated attribute labels are concatenated and fed into the discriminator to match the distribution of real labels.

**Causal Perspective.**  CausalGAN (Kocaoglu et al., 2017) aims to model the full causal generative process of the attribute labels. This involves learning the causal relationships between all pairs of variables. For instance, the causal controller in CausalGAN can be used to measure the influence of ancestors on a given node. However, such an approach demands large amounts of training data to accurately learn these dependencies—a point we validate through experiments.

In contrast, our method focuses only on modeling the causal relationships from parent nodes to a given attribute, which is sufficient for handling the correlations in the data. For example, generating eyeglasses may inadvertently increase the perceived age, even though the true causal direction is age $\rightarrow$ eyeglasses. Our causal conditioning approach is therefore simpler (requiring no separate pretraining) and more data-efficient.

**Empirical Comparison.**  CausalGAN was originally implemented on a smaller causal graph with around 10 variables using the CelebA-64×64 dataset (162,770 images). For a fair comparison in our setting, we re-implemented CausalGAN using the StyleGAN architecture. However, we found that the 70,000 samples available in the FFHQ-512×512 dataset were insufficient to model the complex causal relationships among the 37 attribute labels.

We trained the causal controller for 100 epochs using the recommended WGAN-GP loss. By the end of training, the outputs of the causal controller appeared nearly binary, matching the format of the real data. However, due to the limited training data and stochastic noise introduced by the controller, it often generated combinations of attribute labels that never occurred in the training set. As a result, when these unrealistic labels were used to train the full CausalGAN, the discriminator easily distinguished between real and fake samples, causing training to collapse prematurely. As shown in Fig.10, it collapses quickly.

To fully compare our causal condition with the causal controller, we introduce another version of CausalGAN in our setting. Specifically, we still use real labels during training to avoid failure, i.e., the trained model will be StyleGAN2-ADA. After training, we apply the causal controller to see if we are able to sample some interventional images. And we term this new model as CausalGAN$^+$. We present the results in Fig. 11. We observe that CausalGAN$^+$ introduce unnecessary changes to the images when we only want to change a single attribute.

## E  CAUSAL DISCOVERY

We perform causal discovery on the FFHQ attribute labels and the causal graph is shown in Figure 12. Specifically, we use the PC algorithm (Spirtes et al., 2001) and the BOSS algorithm (Andrews et al., 2023), the latter being designed to efficiently handle large numbers of variables and dense graphs. We first apply the PC algorithm to obtain an initial causal graph, then use BOSS to verify its structure.

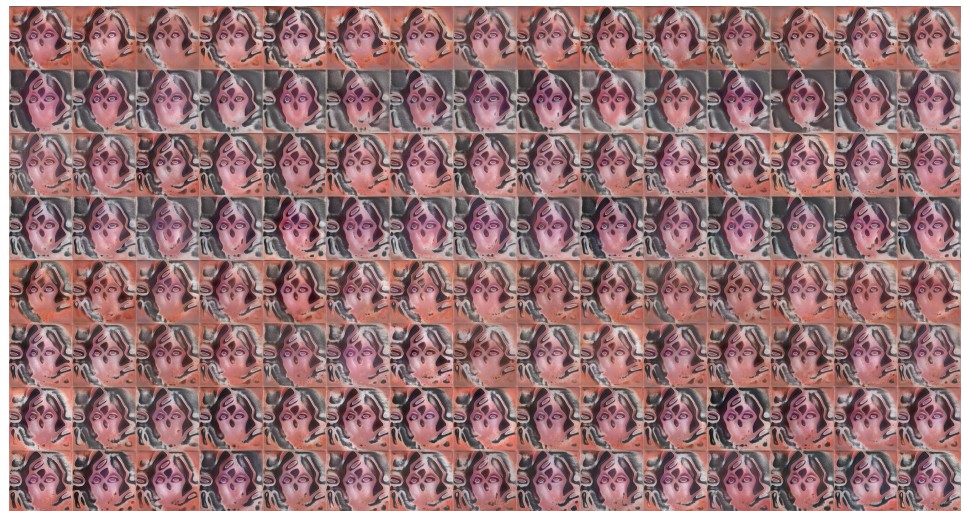

Figure 10: CausalGAN requires a large amount of training data to model complex causal relationships among a greater number of attribute labels. We train its causal controller on the FFHQ dataset, but it fails to fully match the distribution of attribute labels. Moreover, due to the randomness introduced by noise in the causal controller, it generates combinations of labels that never appear in the training data. As a result, CausalGAN quickly fails during the subsequent image generation training.

If both algorithms produce the same edge, we treat it as reliable. In practice, we find that PC outputs generally align well with human domain knowledge. While some edges remain unoriented, our goal is not to develop a new causal discovery algorithm, but simply to obtain enough structural information to guide controllable image generation. In ambiguous cases, we follow established practice in applied causal modeling by incorporating minimal domain knowledge to resolve uncertain directions.

# F PROOF OF IDENTIFIABILITY THEORY

**Theorem 1** (Identifying latent factors with a generative model (**Formal**)). *Consider the data generating process defined in Eq. equation 1 that satisfies the following assumptions:*

- *A1 (Smooth and positive density): The density $p_{\epsilon_i}$ is smooth and positive everywhere.*

- *A2 (Diffeomorphism): The generating function $g$, activation function $\mathbf{f}_i^1$, and deactivating function $\mathbf{f}_i^0$ are $\mathcal{C}^2$-diffeomorphisms onto their corresponding images.*

- *A3 (Latent factor regularity): The activating function $\mathbf{f}_i^1$ differs sufficiently from the deactivating function $\mathbf{f}_i^0$, and each latent factor is of dimensionality one. Specifically, for $i = 1, \ldots, m$ and every $\epsilon_i$, we have*

$$\left( \frac{p_{\epsilon_i}}{(\mathbf{f}_i^1)'} \right)' \neq \left( \frac{p_{\epsilon_i}}{(\mathbf{f}_i^0)'} \right)',$$

*where the derivative $(\cdot)'$ is taken w.r.t. $\epsilon_i$.*

- *A4 (Variability): For each latent factor $\mathbf{z}_i$, there exist two attribute labels, denoted as $\mathbf{T}^{(k)}$ and $\mathbf{T}^{(l)}$, that differ only in the $i$-th entry.*

*Then, for a generative model that assumes the same generative process, satisfies the assumptions above, and matches the data distribution, it identifies the true latent variables $\mathbf{z}_i, i = 1, \ldots, m$ up to an invertible transformation for each attribute $\mathbf{T}_i$.*

*Proof.* To ligthen the notation, let $\mathbf{z} = (\mathbf{z}_c, \mathbf{z}_1, \mathbf{z}_2, \ldots, \mathbf{z}_m)$ and $\tilde{\mathbf{z}} = (\tilde{\mathbf{z}}_c, \tilde{\mathbf{z}}_1, \tilde{\mathbf{z}}_2, \ldots, \tilde{\mathbf{z}}_m)$. We then have $\mathbf{x} = g(\mathbf{z})$ and $\hat{\mathbf{x}} = \tilde{g}(\tilde{\mathbf{z}})$. Applying the change-of-variable formula with matched data

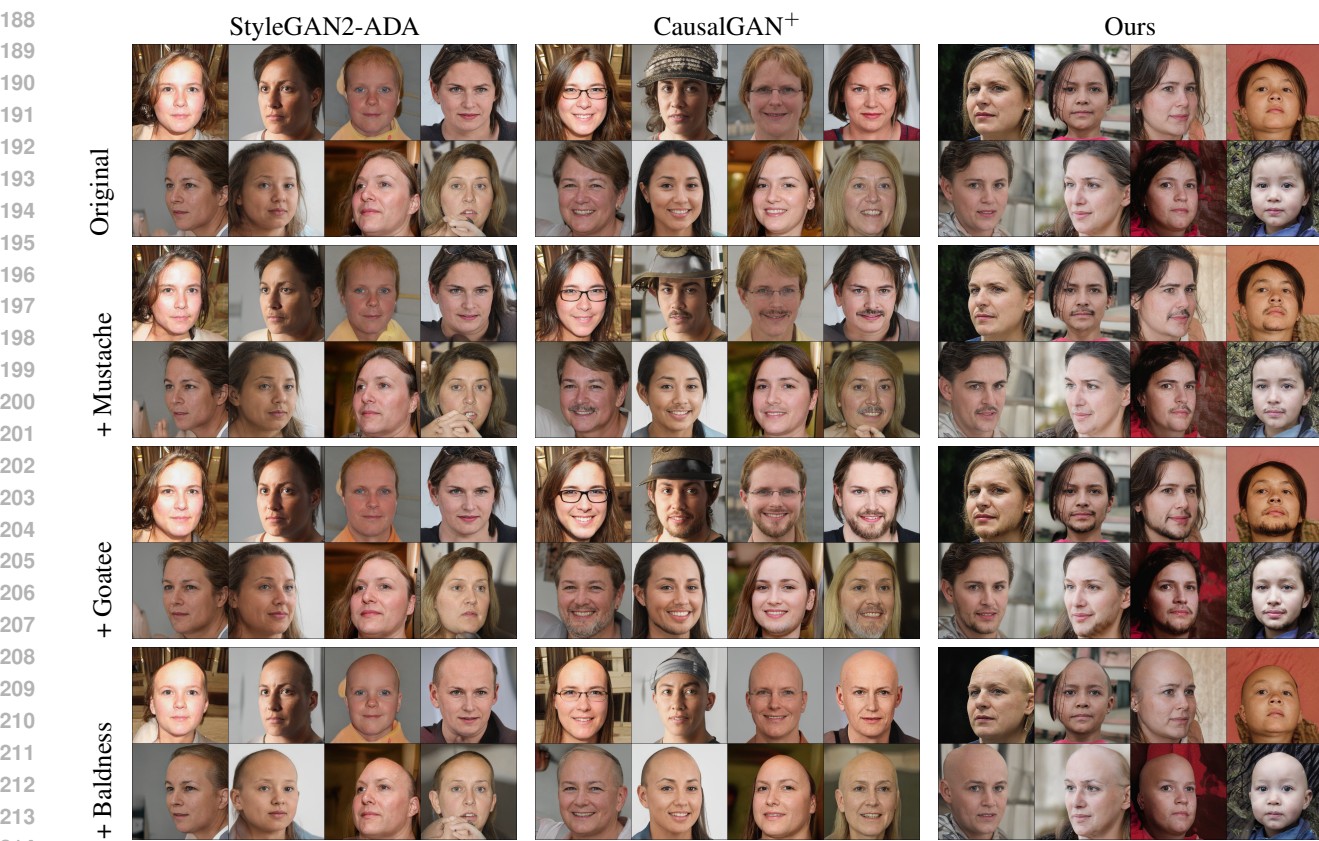

Figure 11: Comparison of controllable image generation. We first fix the random noise and generate girl images. Then we activate the mustache, goatee and baldness concepts for each method. The StyleGAN2-ADA is unable to generate female with mustache images. While the CausalGAN is able to add such attributes, we observe significant unnecessary changes. In contrast, our method is able to change the target attributes only.

distribution yields

$$p_{\hat{\mathbf{x}}} = p_{\mathbf{x}}$$

$$p_{\tilde{g}(\tilde{\mathbf{z}})} = p_{g(\mathbf{z})}$$

$$p_{g^{-1} \circ \tilde{g}(\tilde{\mathbf{z}})} \operatorname{vol} J_{g^{-1}} = p_{\mathbf{z}} \operatorname{vol} J_{g^{-1}}$$

$$p_{h(\tilde{\mathbf{z}})} = p_{\mathbf{z}}.$$

Here, $J_{g^{-1}}$ denotes the Jacobian matrix of function $g^{-1}$ and $h = g^{-1} \circ \tilde{g}$ is the function from $\mathbf{z}$ to $\tilde{\mathbf{z}}$. Applying the change-of-variable formula again, we have

$$p(\tilde{\mathbf{z}}) = p(\mathbf{z}) |\det J_h|.$$

where $J_h$ denotes the Jacobian matrix of function $h$. Since $\tilde{\mathbf{z}}$ and $\mathbf{z}$ consist of independent latent variables, we obtain

$$p(\tilde{\mathbf{z}}_c) \prod_{i=1}^{m} p(\tilde{\mathbf{z}}_i) = p(\mathbf{z}_c) \prod_{i=1}^{m} p(\mathbf{z}_i) |\det J_h|. \tag{10}$$

Consider latent variable $\tilde{\mathbf{z}}_i$ where $i = 1, \ldots, m$. By Assumption A4, there exist two attribute labels, denoted as $\mathbf{T}^{(k)}$ and $\mathbf{T}^{(l)}$, that differ only in the $i$-th entry. Without loss of generality, suppose $\mathbf{T}_i^{(k)} = 1$ and $\mathbf{T}_i^{(l)} = 0$; if this is not the case, we can swap $\mathbf{T}^{(k)}$ and $\mathbf{T}^{(l)}$. Denote the corresponding density functions for these two attribute labels by $p^{(k)}(\tilde{\mathbf{z}}), p^{(k)}(\mathbf{z})$ and $p^{(l)}(\tilde{\mathbf{z}}), p^{(l)}(\mathbf{z})$, respectively. Substituting these density functions into Eq. (10) implies

$$p^{(k)}(\tilde{\mathbf{z}}_c) \prod_{i=1}^{n} p^{(k)}(\tilde{\mathbf{z}}_i) = p^{(k)}(\mathbf{z}_c) \prod_{i=1}^{n} p^{(k)}(\mathbf{z}_i) |\det J_h|. \tag{11}$$

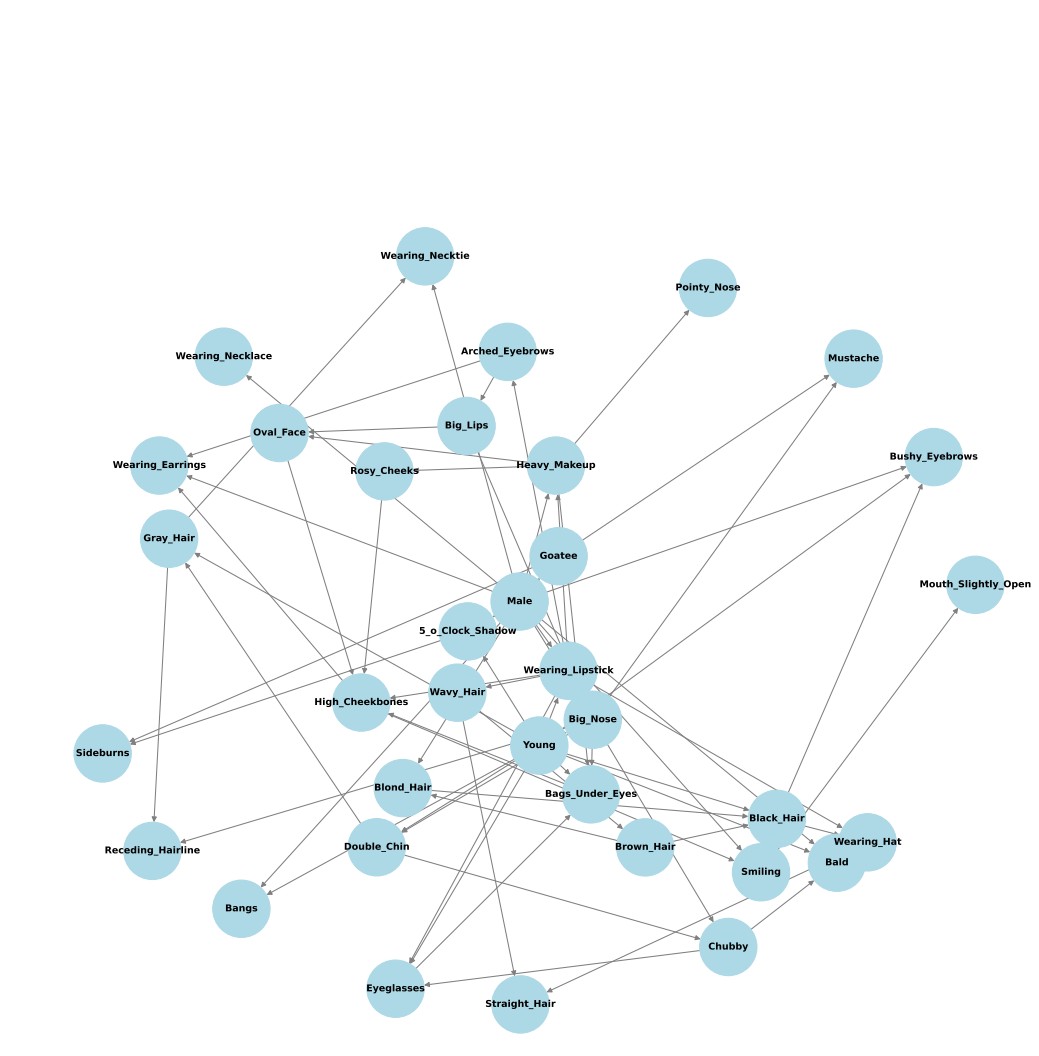

Figure 12: Causal discovery result on the FFHQ attributes labels.

$$p^{(l)}(\tilde{\mathbf{z}}_c) \prod_{i=1}^{n} p^{(l)}(\tilde{\mathbf{z}}_i) = p^{(l)}(\mathbf{z}_c) \prod_{i=1}^{n} p^{(l)}(\mathbf{z}_i) |\det J_h|. \tag{12}$$

Taking quotients of Eqs. (11) and (12), we have

$$\frac{p^{(k)}(\tilde{\mathbf{z}}_i)}{p^{(l)}(\tilde{\mathbf{z}}_i)} = \frac{p^{(k)}(\mathbf{z}_i)}{p^{(l)}(\mathbf{z}_i)}.$$

Here, we used the definition that $p(\tilde{\mathbf{z}}_c)$ and $p(\mathbf{z}_c)$ are invariant across different attribute labels, as well as the assumption that $\mathbf{T}^k$ and $\mathbf{T}^l$ differ only in the $i$-th entry. Now suppose $j \neq i$. Taking first-order derivative w.r.t $\tilde{\mathbf{z}}_j$ in the equation above yields

$$\begin{aligned}
0 &= \frac{\partial}{\partial \tilde{\mathbf{z}}_j} \left( \frac{p^{(k)}(\mathbf{z}_i)}{p^{(l)}(\mathbf{z}_i)} \right) \\
&= \frac{\partial}{\partial \mathbf{z}_i} \left( \frac{p^{(k)}(\mathbf{z}_i)}{p^{(l)}(\mathbf{z}_i)} \right) \frac{\partial \mathbf{z}_i}{\partial \tilde{\mathbf{z}}_j} \\
&= \frac{p^{(l)}(\mathbf{z}_i) \frac{\partial p^{(k)}(\mathbf{z}_i)}{\partial \mathbf{z}_i} - p^{(k)}(\mathbf{z}_i) \frac{\partial p^{(l)}(\mathbf{z}_i)}{\partial \mathbf{z}_i}}{p^{(l)}(\mathbf{z}_i)^2} \frac{\partial \mathbf{z}_i}{\partial \tilde{\mathbf{z}}_j}.
\end{aligned}$$

By Assumption A1, $p^{(l)}(\mathbf{z}_i)^2 \neq 0$. Therefore, we have

$$\begin{aligned}
0 &= \left( p^{(l)}(\mathbf{z}_i) \frac{\partial p^{(k)}(\mathbf{z}_i)}{\partial \mathbf{z}_i} - p^{(k)}(\mathbf{z}_i) \frac{\partial p^{(l)}(\mathbf{z}_i)}{\partial \mathbf{z}_i} \right) \frac{\partial \mathbf{z}_i}{\partial \tilde{\mathbf{z}}_j} \\
&= \left( p(\mathbf{f}_i^0(\epsilon_i)) \frac{\partial p(\mathbf{f}_i^1(\epsilon_i))}{\partial \mathbf{z}_i} \right. \\
&\qquad \left. - p(\mathbf{f}_i^1(\epsilon_i)) \frac{\partial p(\mathbf{f}_i^0(\epsilon_i))}{\partial \mathbf{z}_i} \right) \frac{\partial \mathbf{z}_i}{\partial \tilde{\mathbf{z}}_j} \\
&= \left( p(\epsilon_i) \left| \frac{\partial \mathbf{f}_i^0(\epsilon_i)}{\partial \epsilon_i} \right|^{-1} \frac{\partial p(\mathbf{f}_i^1(\epsilon_i))}{\partial \mathbf{z}_i} \right. \\
&\qquad \left. - p(\epsilon_i) \left| \frac{\partial \mathbf{f}_i^1(\epsilon_i)}{\partial \epsilon_i} \right|^{-1} \frac{\partial p(\mathbf{f}_i^0(\epsilon_i))}{\partial \mathbf{z}_i} \right) \frac{\partial \mathbf{z}_i}{\partial \tilde{\mathbf{z}}_j}.
\end{aligned}$$

Multiply both sides by $\left|\frac{\partial \mathbf{f}_i^1(\epsilon_i)}{\partial \epsilon_i}\right|\left|\frac{\partial \mathbf{f}_i^0(\epsilon_i)}{\partial \epsilon_i}\right|$ yields

$$
\begin{aligned}
0 &= \left(\frac{\partial p(\mathbf{f}_i^1(\epsilon_i))}{\partial \mathbf{z}_i}\left|\frac{\partial \mathbf{f}_i^1(\epsilon_i)}{\partial \epsilon_i}\right|\right. \\
&\quad \left. - \frac{\partial p(\mathbf{f}_i^0(\epsilon_i))}{\partial \mathbf{z}_i}\left|\frac{\partial \mathbf{f}_i^0(\epsilon_i)}{\partial \epsilon_i}\right|\right) p(\epsilon_i)\frac{\partial \mathbf{z}_i}{\partial \tilde{\mathbf{z}}_j} \\
&= \left(\frac{\partial p(\mathbf{f}_i^1(\epsilon_i))}{\partial \epsilon_i}\,\mathrm{sgn}\left(\frac{\partial \mathbf{f}_i^1(\epsilon_i)}{\partial \epsilon_i}\right)\right. \\
&\quad \left. - \frac{\partial p(\mathbf{f}_i^0(\epsilon_i))}{\partial \epsilon_i}\,\mathrm{sgn}\left(\frac{\partial \mathbf{f}_i^0(\epsilon_i)}{\partial \epsilon_i}\right)\right) p(\epsilon_i)\frac{\partial \mathbf{z}_i}{\partial \tilde{\mathbf{z}}_j} \\
&= \left(\frac{\partial}{\partial \epsilon_i}\left(p(\epsilon_i)\left|\frac{\partial \mathbf{f}_i^1(\epsilon_i)}{\partial \epsilon_i}\right|^{-1}\right)\mathrm{sgn}\left(\frac{\partial \mathbf{f}_i^1(\epsilon_i)}{\partial \epsilon_i}\right)\right. \\
&\quad \left. - \frac{\partial}{\partial \epsilon_i}\left(p(\epsilon_i)\left|\frac{\partial \mathbf{f}_i^0(\epsilon_i)}{\partial \epsilon_i}\right|^{-1}\right)\mathrm{sgn}\left(\frac{\partial \mathbf{f}_i^0(\epsilon_i)}{\partial \epsilon_i}\right)\right) \\
&\quad \cdot p(\epsilon_i)\frac{\partial \mathbf{z}_i}{\partial \tilde{\mathbf{z}}_j} \\
&= \left(\frac{\partial}{\partial \epsilon_i}\left(p(\epsilon_i)\left(\frac{\partial \mathbf{f}_i^1(\epsilon_i)}{\partial \epsilon_i}\right)^{-1}\right)\right. \\
&\quad \left. - \frac{\partial}{\partial \epsilon_i}\left(p(\epsilon_i)\left(\frac{\partial \mathbf{f}_i^0(\epsilon_i)}{\partial \epsilon_i}\right)^{-1}\right)\right) p(\epsilon_i)\frac{\partial \mathbf{z}_i}{\partial \tilde{\mathbf{z}}_j} \\
&= \left(\left(\frac{p_{\epsilon_i}}{(\mathbf{f}_i^1)'}\right)' - \left(\frac{p_{\epsilon_i}}{(\mathbf{f}_i^0)'}\right)'\right)p_{\epsilon_i}\frac{\partial \mathbf{z}_i}{\partial \tilde{\mathbf{z}}_j},
\end{aligned}
$$

where $\mathrm{sgn}(\cdot)$ is the sign function and the derivative $(\cdot)'$ is taken w.r.t. $\epsilon_i$. By Assumptions A1 and A3, we have

$$
\left(\left(\frac{p_{\epsilon_i}}{(\mathbf{f}_i^1)'}\right)' - \left(\frac{p_{\epsilon_i}}{(\mathbf{f}_i^0)'}\right)'\right)p_{\epsilon_i} \neq 0,
$$

which implies

$$
\frac{\partial \mathbf{z}_i}{\partial \tilde{\mathbf{z}}_j} = 0.
$$

For each $\mathbf{z}_i$, since we are able to perform the above procedure for each $\tilde{\mathbf{z}}_l$ where $l \neq i$ and $\tilde{\mathbf{z}}_c$, each $\mathbf{z}_i$ is solely a function of $\tilde{\mathbf{z}}_i$, i.e., $\mathbf{z}_i = h_i(\tilde{\mathbf{z}}_i)$. This implies that, the row of the Jacobian matrix $\frac{\partial \mathbf{z}}{\partial \tilde{\mathbf{z}}}$ that corresponds to $\mathbf{z}_i$ has only one nonzero entry for $i = 1, \ldots, m$. Denote by $\tilde{\mathbf{z}}_{[m]} = (\tilde{\mathbf{z}}_1, \ldots, \tilde{\mathbf{z}}_n)$ and $\mathbf{z}_{[m]} = (\mathbf{z}_1, \ldots, \mathbf{z}_n)$. The above derivation indicates $\frac{\partial \mathbf{z}_{[m]}}{\partial \tilde{\mathbf{z}}_c} = 0$. Since $h$ is a diffeomorphism and $\frac{\partial \mathbf{z}}{\partial \tilde{\mathbf{z}}}$ is of full rank, the matrix $\frac{\partial \mathbf{z}_{[m]}}{\partial \tilde{\mathbf{z}}_{[m]}}$ must also be of full row rank because $\frac{\partial \mathbf{z}_{[m]}}{\partial \tilde{\mathbf{z}}_c} = 0$. This indicates that $h_i$ is invertible, i.e., $\tilde{\mathbf{z}}_i = h_i^{-1}(\mathbf{z}_i)$. Therefore, $\tilde{\mathbf{z}}_i$ is solely a function of $\mathbf{z}_i$. $\qquad\square$

## G  IMPLEMENTATION DETAILS

We provide the training code in the supplementary material. Our method is built upon StyleGAN2-ADA (Karras et al., 2020), with our main empirical contribution focused on a redesign of the mapping network used in StyleGAN2-ADA. In traditional StyleGAN-based approaches, Gaussian noise and a class embedding are jointly processed by an MLP to produce an entangled latent representation in the $\mathcal{W}$ space.

For each attribute $A_i$, we employ a two-layer MLP to transform an input noise vector $\epsilon_i$ into an activated or deactivated concept representation. These outputs are then concatenated to form a latent vector $\mathbf{z}$, yielding a new latent space $\mathcal{Z}$ instead of the conventional $\mathcal{W}$ space.

Ours (w/o sparsity)      Ours (w/o causal)      Ours

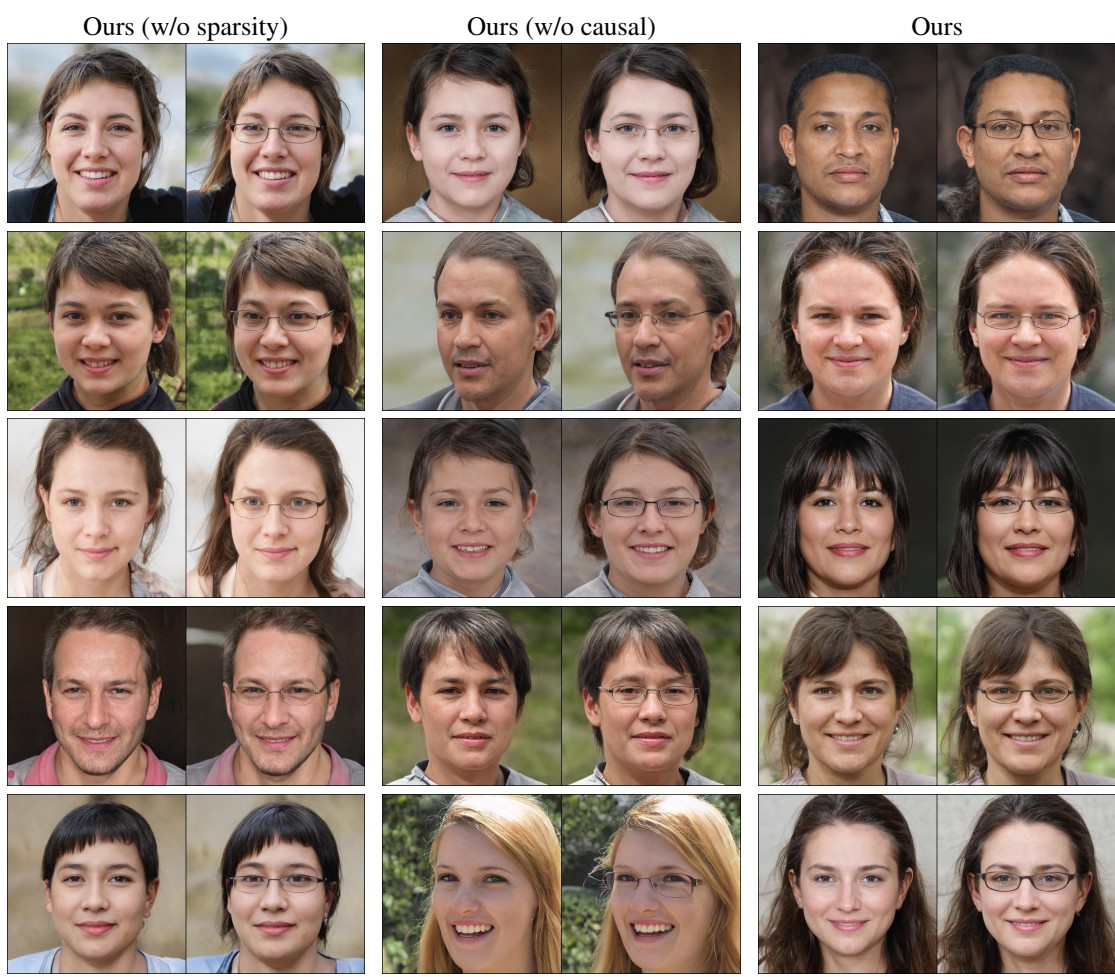

Figure 13: **Qualitative Ablation Results**. Without sparsity and causal modeling, the output images suffer more distortions when we want to add eyeglasses only.

At the beginning of training, we set the dimensionality of each attribute representation to 20. Additionally, we introduce a learnable mask $m_i$ for each attribute to allow the model to select the relevant dimensions for each concept. To promote sparsity, we apply an $L_1$ regularization on the mask, typically using a sparsity weight of $\lambda_{\text{sparsity}} = 0.1$.

## H    MORE ABLATION RESULTS

We present the qualitative ablation results in Fig. 13. Without sparsity, the generated outputs exhibit noticeable distortions, such as changes in hairstyle and apparent age. Adding sparsity regularization mitigates these issues by reducing unintended attribute alterations. However, due to the strong dependency between age and eyeglasses, adding eyeglasses still results in an older appearance, highlighting the necessity of causal modeling. In contrast, our full method successfully generates eyeglasses without affecting other attributes in the output image. We also report the quantitative results in Table 5.

Incorporating sparsity improves the similarity score by discouraging changes to unrelated attributes. Nevertheless, due to the strong correlation between eyeglasses and age, fully disentangling these attributes remains challenging. In contrast, introducing the causal modeling module leads to a substantial improvement in identity similarity, highlighting the importance of causal

| Method | Eyeglasses | |
|---|---|---|
| | DINO ↑ | ID ↑ |
| Ours, W/O Sparsity | 0.935 | 0.730 |
| Ours, W/O Causal | 0.943 | 0.734 |
| Ours | **0.953** | **0.775** |

Table 5: **Ablation Results.** Sparsity helps preserve unrelated attributes, while causal modeling enhances identity preservation.

modeling for faithful and controlled attribute manipulation.

