# OpenReview forum: "Attribute-Guided Image Generation with Causally Disentangled Representation"
_ICLR.cc/2026/Conference — ICLR 2026 Conference Withdrawn Submission_

### Official Review · Reviewer_v3BA · 2025-10-30

**Soundness:** 2
**Presentation:** 2
**Contribution:** 2
**Rating:** 4
**Confidence:** 3

**Summary:**

Previous methods for controlled generation suffer from poor attribute disentanglement and unwanted attribute correlations. To solve these problems, the paper proposes a novel attribute-guided generative framework. Main contributions are as follows:

1.  **Mask-Based Representation Learning:** A technique that learns a mask-based representation for each attribute label and ensures that each attribute label influences only a minimal subset of the representation dimensions.
2.  **Causal Conditioning Strategy:** This strategy learns attribute-specific representations conditioned only on their causal parents. This strategy reduces undesirable correlations.

**Strengths:**

-   **[S1] Novel Approach:** Combining a novel mask-based representation for disentanglement with a causal conditioning strategy for correlation reduction is novel. Further, the authors provide a theoretical proof in Sec 3.3 to solidify their claims.

**Weaknesses:**

-   **[W1] Lack of a standardized qualitative comparison for the attribute editing tasks:** In Fig. 4,6,7 and 11, the proposed method and the baselines are applied to different input images. This  does not seem to be a fair to fair comparison as it makes difficult to assess the relative performance and failure modes of the models.

-   **[W2] Comparison with recent models like Flux Kontext:** Given that this model is known to work on _open-ended_ conditional generation, a direct comparison is required to assess the proposed method's claim on the recent generative methods.
-   **[W3] Difficult to extend to diffusion/flow-based generative models:** The paper is based on StyleGAN, whose latent space (W space) is disentangled and has semantic attributes. In comparison, recent diffusion-based models do not have this W space and conditions using a cross-attention mechanism. Furthermore, the causal conditioning method will not extend to the diffusion-based generative methods.

**Questions:**

-   [Q1] Have authors tried other attributes like lighting conditions, pose variations etc?
-   [Q2] How about other correlated attributes like "age and hair"?

---

### Official Review · Reviewer_FnRx · 2025-10-31

**Soundness:** 3
**Presentation:** 3
**Contribution:** 2
**Rating:** 4
**Confidence:** 4

**Summary:**

This paper presents a novel framework for controllable image generation that integrates causal representation learning with attribute-guided modeling. The authors identify two long-standing challenges in this field: (1) achieving disentanglement of attribute representations, and (2) mitigating spurious correlations between attributes (e.g., “eyeglasses” co-varying with “age”). To address these, they introduce a mask-based representation learning scheme and a causal conditioning mechanism.

The mask-based component enforces sparsity—each attribute affects only a small subset of the latent representation—thus promoting disentanglement and interpretability. The causal conditioning strategy explicitly models inter-attribute dependencies using classical causal discovery methods (e.g., PC algorithm) and intervenes on exogenous variables to decorrelate attributes during generation. The paper further provides theoretical identifiability guarantees for recovering latent factors associated with individual attributes.

Empirically, the proposed model is implemented on the StyleGAN2-ADA backbone and evaluated across diverse datasets including AFHQDog, ZAPPOS, ColorMNIST, LSUN Bedroom, and FFHQ. Quantitative metrics such as FID, ID, DINO similarity, and a custom disentanglement score consistently demonstrate improvements over baselines (StyleGAN2-ADA, InterfaceGAN, WPlus, ConceptSlider, SANA, and Flux 1-dev). Visual comparisons show that the method achieves fine-grained, smooth control of target attributes without unintended modifications.

The paper also discusses extensions to text-to-image models (Flux 1-dev) and demonstrates promising improvements through time-based masking and causal conditioning. Theoretical rigor, strong experimental results, and clear motivation establish this work as a substantial contribution toward interpretable and causally robust controllable generation.

**Strengths:**

1. Conceptual novelty - The integration of causal discovery with attribute-guided image generation is elegant and timely. While prior works (InterfaceGAN, StyleCLIP) rely on linear or heuristic disentanglement, this paper introduces a principled causal framework backed by identifiability proofs.

2. Technical rigor - The paper provides formal theoretical analysis (Theorem 1) guaranteeing recovery of attribute-specific latent factors, which is rarely attempted in generative modeling papers. The causal conditioning process is mathematically grounded in structural causal models, giving a firm foundation to the proposed methodology.

3. Comprehensive experiments -The authors benchmark across both independent and correlated-attribute datasets with diverse visual domains. Quantitative gains in FID and disentanglement, alongside extensive visual results, convincingly support the claims. Ablation studies (with/without mask, with/without causal module) are thoughtfully designed and reveal clear contributions of each component.

4. Clarity and organization – The paper is well-written and well-structured. Figures clearly illustrate mechanisms and qualitative effects. The authors explicitly contrast their work against CausalGAN, CausalVAE, and ConceptSlider, situating it firmly within the literature.

**Weaknesses:**

1. Computational limitations - The paper restricts experiments to GANs, noting diffusion models are “computationally prohibitive.” Given the field’s rapid shift toward diffusion-based architectures, this may limit immediate practical relevance.

2. Empirical scope of causal discovery - The causal structure learning is briefly described (using PC algorithm and minimal prior knowledge). However, the reliability and stability of the learned causal graph across datasets are not deeply analyzed. How sensitive are results to errors in causal edge orientation?

3. Limited real-world complexity - The datasets used (AFHQ, ZAPPOS, FFHQ) have relatively simple and well-annotated attributes. It remains unclear how the method performs when attributes are continuous, hierarchical, or ambiguous (e.g., aesthetic style, emotion intensity).

4. Comparative baselines – The strongest diffusion-based controllable generation methods (e.g., Uni-ControlNet, T2I-Adapter, InstructPix2Pix) are omitted. While computationally expensive, referencing their expected differences would clarify the paper’s relative positioning.

**Questions:**

1. Causal Graph Learning: How robust is your method to mis-specified causal graphs? If the PC algorithm yields incorrect edges, does the model still retain disentanglement due to the mask mechanism, or do performance metrics degrade sharply?

2. Scalability: Could the causal conditioning mechanism scale to high-dimensional attribute spaces (e.g., > 50 labels as in CelebA-HQ)? Does the computational cost of discovering and conditioning on the causal structure grow quadratically with attribute count?

3. Continuous Attributes: Your framework seems tailored for binary labels. How would it extend to continuous or ordinal attributes (e.g., “age,” “smile intensity”)? Would the mask formulation or identifiability proof still hold?

4. Attribute Interaction Visualization: It would be insightful to include quantitative or graphical analysis of learned causal effects (e.g., how much “age” changes when “eyeglasses” is toggled). Are such diagnostics available?

---

### Official Review · Reviewer_u7aR · 2025-11-01

**Soundness:** 2
**Presentation:** 3
**Contribution:** 2
**Rating:** 2
**Confidence:** 2

**Summary:**

The paper proposes a controllable image generation framework that learns mask-based attribute representations to achieve disentanglement and applies causal conditioning to reduce spurious correlations between attributes. The model uses an attribute-invariant code and attribute-specific codes, then mixes them via learnable masks before feeding into the generator. Experiments on benchmarks show improved FID and stronger edit controllability v.s. GAN baselines and other text-to-image methods.

**Strengths:**

1. The paper has a clear problem focus, with disentanglement + correction, which is reasonable and classical setting for causal representation learning.

2. The mask mixing of attribute-specific and attribute-invariant latents is pretty intuitive and elegant.

3. The experiments have shown the effectiveness of the method.

**Weaknesses:**

1. The theorem 1 is stated pretty informally and I think the invertible mapping assumption might be too strong. I would suggest the author to add more explanation to this part. It would also be great if author could empirically check the assumption by, e.g., checking mutual information between latent variables and factors.

2. In Eq. (2), without any constraint on z_c or regularizations, I don't think z_c is irrelevant to any features. Leak of attribute information to z_c must harm the disentanglement.

3. Sparsity on each m_i cannot secure mutual exclusivity over attributes. The author should explain more here or do some ablation studies.

4. The evaluation process lacks diffusion baselines. Without training and evaluation on diffusion baselines under the same settings, it's hard to tell if the performance gain comes from the proposed method, or simple from difference between GANs and diffusion models.

**Questions:**

Please refer to my comments in weaknesses.

---

### Official Review · Reviewer_vkww · 2025-11-05

**Soundness:** 3
**Presentation:** 3
**Contribution:** 3
**Rating:** 6
**Confidence:** 2

**Summary:**

The paper studies attribute-guided image generation with two main pieces: (1) per-attribute latents gated by a learnable sparse mask so each attribute only touches a tiny slice of the representation, and (2) a causality-aware conditioning step that uses parent-conditioned surrogates so edits to a child label don’t “leak” into its parents. There’s also an identifiability argument tailored to this setup. Empirically, the method shows better controllability and competitive image quality across several datasets, plus cleaner edits on correlated face attributes; ablations support that both the sparsity and causal parts matter.

**Strengths:**

The masking idea is simple. The implementation seems straightforward to reproduce.

The causal conditioning is lightweight compared to training a full causal label generator, yet seems to reduce side effects in edits.

Results look consistent with the story.

**Weaknesses:**

Robustness to mistakes in the discovered causal graph isn’t really explored; it’s unclear how sensitive the method is if some edges are wrong or missing.

The identifiability claim relies on assumptions that are hard to verify with noisy, correlated labels; practical consequences of violations aren’t discussed.

**Questions:**

Robustness to causal-graph errors isn’t really explored; it’s not obvious how sensitive the method is if some edges are wrong or missing.

The identifiability claim rests on assumptions that are not testable. Practical consequences of violations aren’t discussed in depth.

Some datasets use automated annotations or classifiers; a clearer read on label-noise sensitivity would help.

---

### Note · Authors · 2025-11-14

I have read and agree with the venue's withdrawal policy on behalf of myself and my co-authors.